# Synchronized excitability in a network enables generation of internal neuronal sequences

Yingxue Wang[1], Zachary Roth[1,2], Eva Pastalkova[1]*

[1]Janelia Farm Research Campus, Howard Hughes Medical Institute, Ashburn, United States; [2]Department of Mathematics, University of Nebraska-Lincoln, Lincoln, United States

**Abstract** Hippocampal place field sequences are supported by sensory cues and network internal mechanisms. In contrast, sharp-wave (SPW) sequences, theta sequences, and episode field sequences are internally generated. The relationship of these sequences to memory is unclear. SPW sequences have been shown to support learning and have been assumed to also support episodic memory. Conversely, we demonstrate these SPW sequences were present in trained rats even after episodic memory was impaired and after other internal sequences – episode field and theta sequences – were eliminated. SPW sequences did not support memory despite continuing to 'replay' all task-related sequences – place- field and episode field sequences. Sequence replay occurred selectively during synchronous increases of population excitability – SPWs. Similarly, theta sequences depended on the presence of repeated synchronized waves of excitability – theta oscillations. Thus, we suggest that either intermittent or rhythmic synchronized changes of excitability trigger sequential firing of neurons, which in turn supports learning and/or memory.

*For correspondence: pastak@janelia.hhmi.org

**Competing interests:** The authors declare that no competing interests exist.

## Introduction

Sequences generated in the absence of sensory cues have been observed in various cortical areas, basal ganglia, and the hippocampus (*Wilson and McNaughton, 1994*; *Nádasdy et al., 1999*; *Louie and Wilson, 2001*; *Dragoi and Buzsáki, 2006*; *O'Neil et al., 2006*; *Foster and Wilson, 2006*; *Ji and Wilson, 2007*; *Lee and Wilson, 2002*; *Pastalkova et al., 2008*; *Luczak et al., 2009*; *Peyrache et al., 2009*; *Havenith et al., 2011*; *Harvey et al., 2012*; *Xu et al., 2012*; *Carrillo-Reid et al., 2015*; *Markowitz et al., 2015*; *Mello et al., 2015*). These internally generated sequences seem to support mental functions such as cognitive planning, motor planning, visual memory, and episodic memory. It has been suggested that the composition of these internal sequences reflects the synaptic connectivity of the network, first, because the formation of internal sequences frequently requires learning (*Gill et al., 2011*; *Xu et al., 2012*) and, second, because similar sequences can be repeated in the absence of prominent sensory cues (*Wilson and McNaughton, 1994*; *Ji and Wilson, 2007*; *Pastalkova et al., 2008*; *Carrillo-Reid et al., 2015*; *Markowitz et al., 2015*).

The hippocampus generates internal sequences that play out over the timescale of tens of milliseconds. Theta sequences are generated during running (*Skaggs et al., 1996*; *Dragoi and Buzsáki, 2006*), and sharp-wave (SPW) sequences are generated during pauses between runs and during sleep (*Nádasdy et al., 1999*; *Ji and Wilson, 2007*). The hippocampus also produces seconds-long internal sequences during running – episode field sequences (*Pastalkova et al., 2008*). Episode field sequences might appear to be like place field sequences (*O'Keefe and Dostrovsky, 1971*); but unlike place field sequences, episode field sequences are formed independently of sensory cues and only during memory tasks (*Wang et al., 2015*).

Each of these internal sequences was generally accepted as being necessary for learning and/or episodic memory tasks such as delayed left-right alternation or radial arm maze tasks. For example, elimination of SPWs impaired learning of these tasks (*Girardeau et al., 2009*; *Dupret et al., 2010*; *Ego-Stengel and Wilson, 2010*; *Jadhav et al., 2012*) and elimination of theta sequences and episode field sequences was accompanied by learning impairment as well as the loss of episodic memory in well-trained animals (*Robbe et al., 2006*; *Wang et al., 2015*).

How do internal sequences support learning? One hypothesis suggests that the waking experience of animals is reactivated during SPW events and that this reactivation strengthens connections within the hippocampal network and/or transfers the newly acquired information into the cortex (*Buzsáki, 1986*). Supportive of this hypothesis, place cells tend to fire during SPWs in an order that is similar to the order in which place fields of the same cells are organized in space, a phenomenon called 'replay' of waking experience (*Wilson and McNaughton, 1994*; *Nádasdy et al., 1999*; *Foster and Wilson, 2006*). Similarly, it has been suggested that the fast succession of firing during theta sequences might trigger synaptic plasticity (*Skaggs et al., 1996*).

An important unresolved question, however, is whether SPW sequences, like theta sequences, support not only the learning of a new task but also the performance of a previously learned episodic memory task. To address this question, we first characterized the replay of SPW sequences during an already learned episodic memory task – a delayed left-right alternation task – and then took advantage of the fact that medial septum (MS) inactivation strongly impairs episodic memory (*Chrobak et al., 1989*; *Mizumori et al., 1990*) but leaves the local-field potential SPW events intact (*Buzsáki, 1984*). We asked whether SPW sequences were preserved during SPW events once episodic memory was eliminated.

First, we show that under normal conditions SPW replay was not limited to sequences that were accompanied by sensory cues, that is, to place field sequences. Rather, SPW sequences also replayed episode field sequences. Place and episode cell sequences were replayed with comparable frequency and their replay was not biased by the current position of the animal or by the previous or future choice of the maze arm. Next, we show that SPW replay of both place field and episode field sequences was preserved after MS inactivation, suggesting that SPW sequences could not support episodic memory of animals after other internal sequences such as theta and episode field sequences were eliminated. Importantly, this data demonstrates that the replay of episode field sequences during SPWs was preserved even after the episode field sequences themselves were eliminated during wheel running. Thus, we suggest that SPW sequences reflect a preserved synaptic structure of the network rather than the memory of recent events.

The fact that despite their similarity to theta and episode field sequences, only SPW sequences survived after MS inactivation made us wonder whether a specific mechanism is required to activate sequences in a network, a mechanism that perhaps remained uninterrupted during SPWs. We show that replay of place field and episode field sequences were generated exclusively during SPW events, during the short periods when the excitability of a large population of neurons suddenly increased. Interestingly, one of the effects of the MS inactivation was elimination of such synchronized changes of excitability during running (i.e., elimination of theta oscillations). Thus, we hypothesize that these isolated or repetitive waves of synchronously rising and decreasing excitability among a large population of neurons are necessary to activate internally generated sequences.

## Results

### Internal hippocampal sequences are replayed during sharp-waves

SPW sequences are thought to play an important role in learning (*Buzsáki, 1989*, *2015*), in part, because neuronal sequences with an order highly similar to place field sequences are replayed during SPWs. First we asked whether episode field sequences, which are internally generated and do not rely on external cues, are also replayed during SPWs.

We analyzed the neuronal activity and local field potentials recorded in the hippocampal CA1 field of three rats performing a delayed alternation task (*Figure 1A* top, *Carr, 1917*; *Tolman, 1925*; *Mizumori et al., 1990*; *Ainge et al., 2007*; *Wang et al., 2015*, *2016*). We identified episode field sequences generated during wheel running (*Figure 1A*-purple, *Pastalkova et al., 2008*) and place field sequences in the arms of the maze (*Figure 1A*-green). We also identified all the SPW events

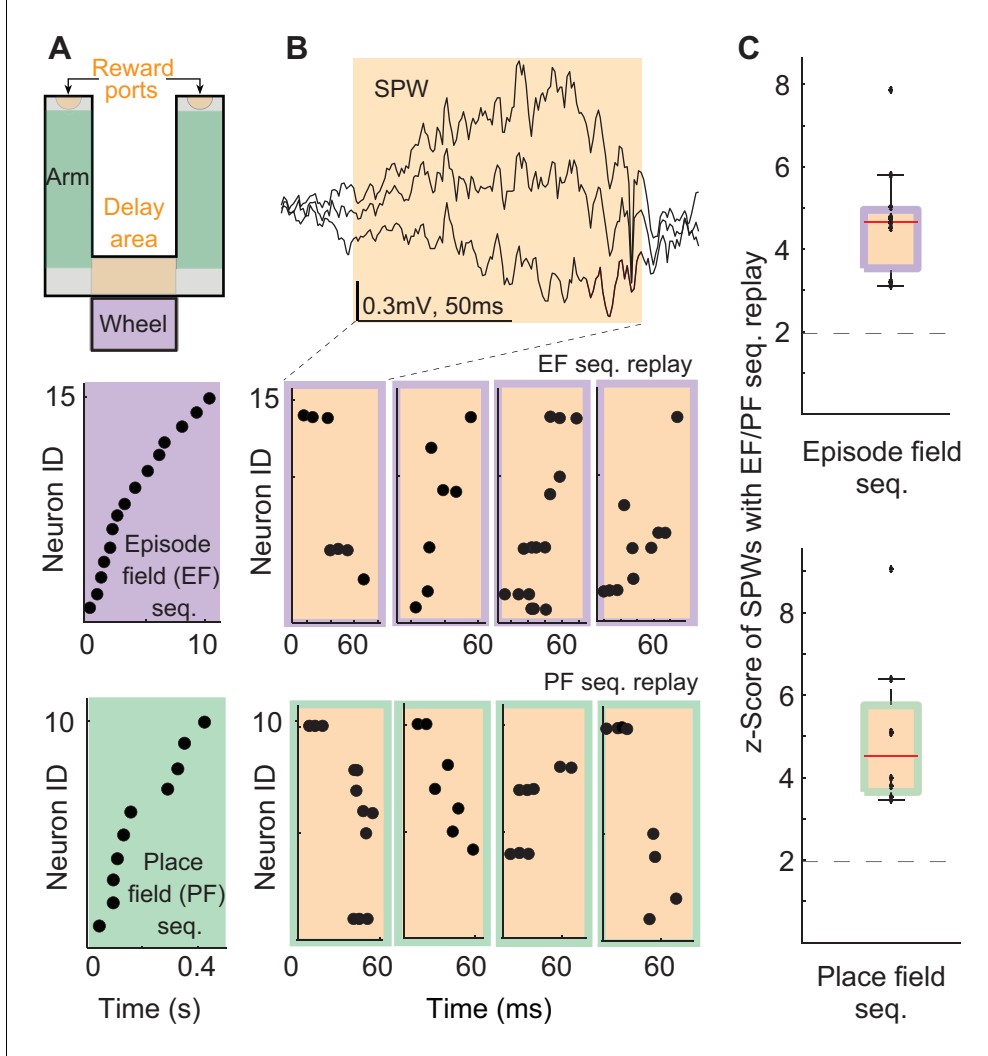

**Figure 1.** Episode field and place field sequence replay during sharp-waves (SPWs). (**A**) An alternation task maze. Purple: an example episode field sequence. Green: an example place field sequence. Each dot represents the peak of an episode/place field. (**B**) An example SPW event. Orange/purple: example SPW sequences correlated with the example episode field sequence. Orange/green: example SPW sequences correlated with the example place field sequence. Each dot represents a spike. (**C**) Percentage of SPWs with sequences significantly correlated with episode field (top) and place field (bottom) sequences. Red line: median, error bars: 95%-tile. Dash line: 2SD. DOI: 10.7554/eLife.20697.002

(*Figure 1B* top) generated in the reward and delay areas of the maze (*Figure 1A* – orange, speed < 3 cm/s; *Csicsvari et al., 2000*). As shown before, 5–20% of SPW sequences were highly correlated with the sensory-cue-guided place field sequences identified in the arms of the maze (*Figure 1B,C* – orange/green, median = 9%; z-score= 3.5–9; median z-score= 4.5; *Ji and Wilson, 2007*; *Foster and Wilson, 2006*; *Diba and Buzsáki, 2007*). Interestingly, about 4.3–16.7% of SPW sequences were also significantly correlated with the episode field sequences (*Figure 1B,C* – orange/purple; median = 10.25%; z-score= 3.1–7.8, median z-score= 4.7, p<0.01). SPW sequences correlated with place or episode field sequences did not show any spatial bias, suggesting they were not strongly influenced by recently traveled trajectories in the maze. (44.8 ± 2.9% of all SPW- sequences significantly correlated with the place field sequence in the left maze arm were generated in the left reward area . 51.2 ± 4% of all SPW sequences significantly correlated with the place field sequence in the right maze arm were generated in the left reward area , n = 8 recordings from 3 animals.) The percentage of significant sequences also was not correlated with the performance of animals (Pearson

correlation: Pre: r=0.34, p=0.25). This data shows that internal sequences (i.e. episode field sequences) are replayed during SPW events with equal probability as sequences strongly influenced by sensory cues (i.e. place field sequences). This suggests that SPW sequences reveal synaptic structure underlying not only place field sequences but also episode field sequences. Also, this data suggests that SPW sequences do not clearly reflect an animal's most recent choices and experiences.

## SPW sequences were present even after memory loss

Since SPW sequences did not directly reflect recent experiences, we asked whether SPW replay could support episodic memory of animals even after internal sequences such as theta and episode field sequences were eliminated (*Wang et al., 2015*). First, we checked the local field potential to determine whether the hippocampal network was able to generate SPW events even after the MS inputs into hippocampus were inhibited (*Buzsáki, 1984*). SPW events were reliably generated (*Figure 2A*) during reward consumption and during short periods of rest in the delay area of the maze (*Figure 1A*-orange, 2B). After the inactivation, SPW events occurred more frequently (*Figure 2C*, Kruskal-Wallis, p=5.7 * 10 e-5, n = 13), recruited more neurons (*Figure 2D*, KS-test = 0.15; p=1.7 e-23; n(Pre) = 1114; n(Musc) = 3997), and tended to have more spikes per neuron (*Figure 2E*, KS-test = 0.11; p=8.4 e-10; n(Pre) = 1114; n(Musc) = 3997) even though the duration of the SPW events did not change (*Figure 2F*, KS-test = 0.08; p>> 0.05). Thus, SPW events were more frequent and stronger after MS inactivation.

Was replay of place and episode field sequences still present after MS inactivation even though the performance of animals in the task was degraded (*Chrobak et al., 1989*; *Mizumori et al., 1990*; *Wange et al., 2015*)? We compared the order of episode field sequences from before the inactivation(*Figure 3A*, left - purple) with the order of SPW sequences from after the inactivation (*Figure 3A*, right - blue/purple). We found that SPW sequences whose orders were highly similar to the pre-inactivation episode field sequences were not eliminated by the inactivation. In fact, after inactivation, a slightly higher percentage of eligible SPW sequences was significantly correlated with episode field sequences (*Figure 3B;* n = 10 recordings; z-score(Pre) = 4.1 ± 0.4, z-score(Musc) = 6.7 ± 0.9; Kruskal-Wallis, p=0.023). The same held true for the SPW sequences correlated with place field sequences (*Figure 3E;* n = 8 recordings; z-score(Pre) = 5 ± 0.7; z-score(Musc) = 8.5 ± 0.9; Kruskal-Wallis, p=0.016). Both place and episode field sequences were replayed at close to a constant rate during the entire recording following MS inactivation (*Figure 3C,F*), and the mean number of neurons activated during the replay of the place and episode field sequences did not change after the inactivation (Episode cells: Pre: 8.17 neurons, Musc: 8.73 neurons; Place cells: Pre: 5.96 neurons, Musc: 5.28 neurons). This was the case even though the episode field sequences observed during wheel running were eliminated after MS inactivation (*Wang et al., 2015*). Correspondingly, there was no significant correlation between the frequency of replay occurrence and the performance of animals (*Figure 4A*, Pre: r = 0.34 p=0.25, Musc: r = −0.19 p=0.53). Also, we did not observe any significant decrease in the sequence replay during error trials (*Figure 4B*, Correct trials: Pre: 0.22 Hz ± 0.014, Musc: 0.32 Hz ± 0.024. Error trials: Pre: 0.5 Hz ± 0.08, Musc: 0.85 Hz ± 0.25). As before the injection, SPW sequences that were correlated with pre-inactivation place and episode field sequences did not show clear spatial bias, suggesting that they were not strongly influenced by recently traveled trajectories in the maze. (56 ± 6.6% of all SPW sequences significantly correlated with the place field sequence in the left maze arm were generated in the left reward area. 63 ± 8.4% of all SPW sequences significantly correlated with the place field sequence in the right maze- arm were generated in the left reward area, n = 8 recordings from 3 animals). Thus, this data further confirms that SPW replay of episode and place field sequences does not support episodic memory in trained animals (*Wang et al., 2015*) and likely reflects the structure of the network rather than the recent activity in the network.

## Synchronized wave of excitability in a population of neurons is necessary for the generation of internal sequences

After MS inactivation, internal sequences (e.g., episode field and theta sequences) were eliminated during running, i.e., during theta brain state (*Wang et al., 2015*). Conversely, highly similar sequences formed by physically identical neurons were preserved during reward consumption, i.e., during SPW brain state. What was the reason one set of these sequences was affected by MS inactivation

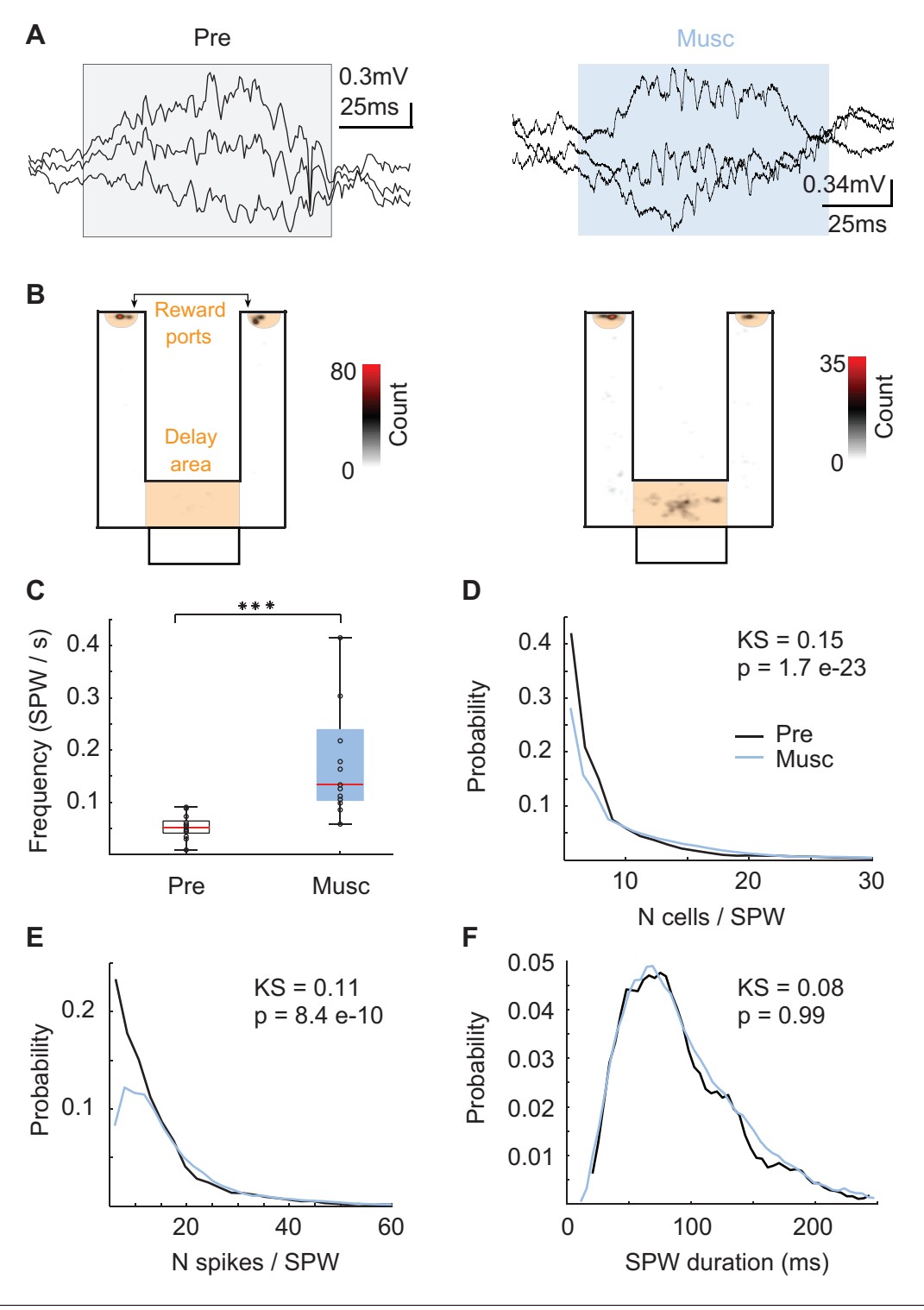

**Figure 2.** Characteristics of SPWs before and after medial septum (MS) inactivation. (**A**) Example SPW event before (black) and after (blue) the inactivation. (**B**) Locations of SPW events before (left) and after (right) the inactivation. (**C**) Frequency of SPW events before (Pre) and after (Musc) the inactivation. (**D**) Number of active neurons during SPW events. (**E**) Number of spikes during SPW events. (**F**) Duration of SPW events.

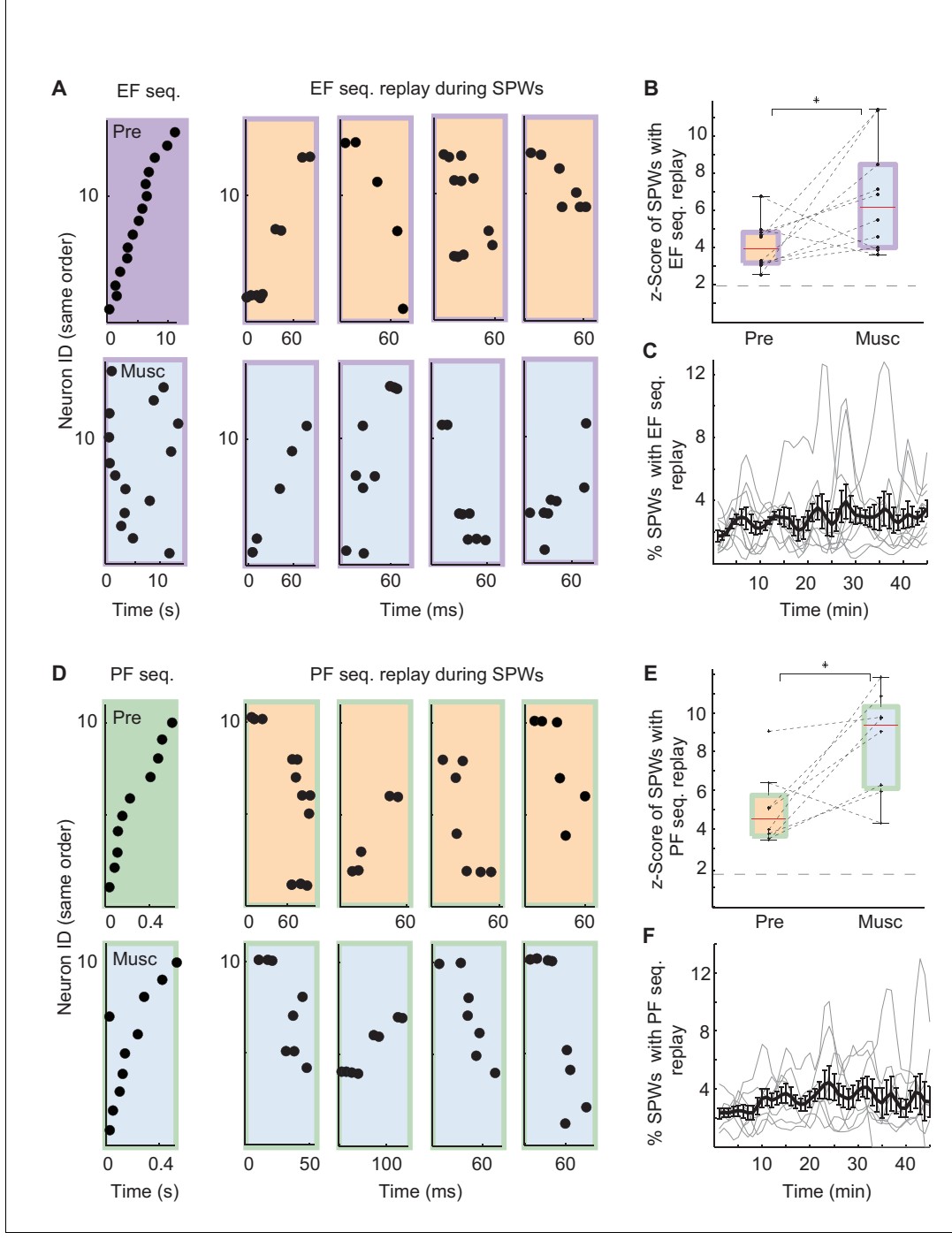

**Figure 3.** SPW replay before and after MS inactivation. (**A**) Left: Example episode field sequence before (purple) and after (blue/purple) MS inactivation. Right: example SPW sequences correlated with the pre-injection episode field sequence recorded before (orange/purple) and after (blue/purple) MS inactivation. (**B**) Normalized percentage of SPW events with episode field replay before (Pre) and after (Musc) MS inactivation. Dash line: 2SD. (**C**) Percentage of SPW events with episode field replay in time since the start of the recording. (**D–F**) the same as **A–C** but for place field sequences.

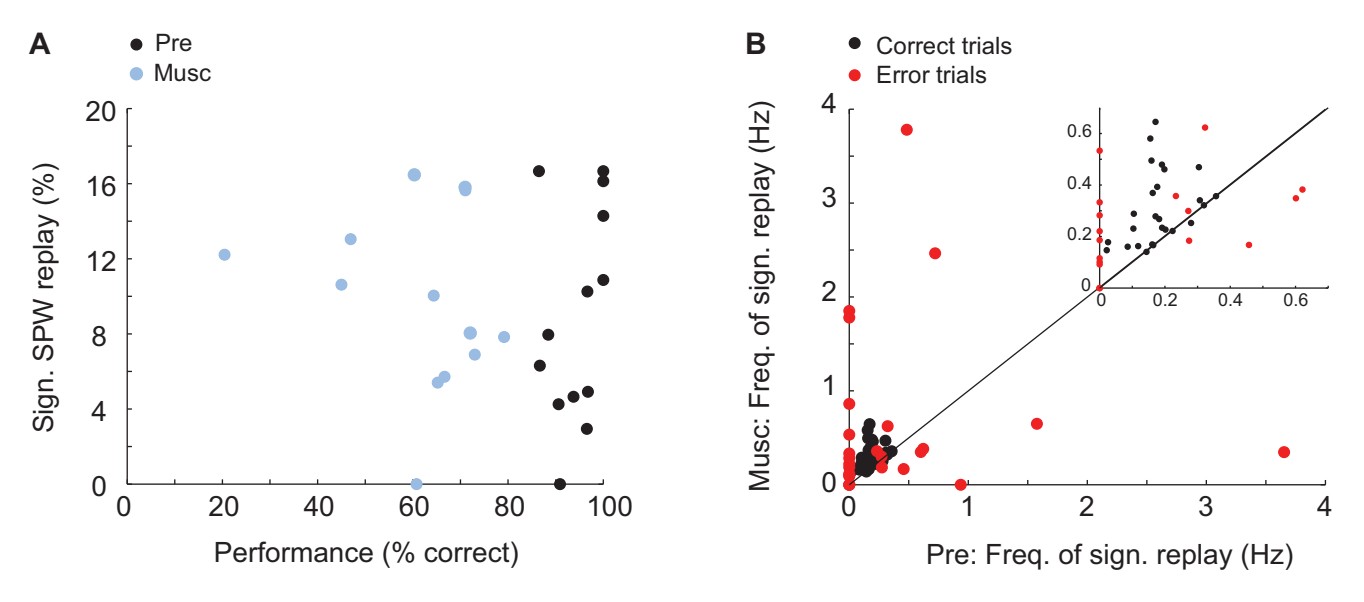

**Figure 4.** Relationship between memory performance and SPW replay. (A) Relationship between animal performance in the task and percentage of significant sequence replay before (Pre) and after (Musc) MS injection. (B) Frequency of sequence replay before (x-axis) and after (y-axis) MS injection plotted separately for correct (red) and erroneous (red) trials. Each dot represents the mean of correct/erroneous trials from one recording.

while the other one remained unchanged? One common feature of the observed internal sequences (i.e., theta and SPW sequences) is that they coincide with synchronized changes in the excitability of a large population of neurons: individual troughs of theta oscillations and SPW events (*Figure 5A* – red traces, 4B-C; *Nádasdy et al., 1999*; *Louie and Wilson, 2001*; *Dragoi and Buzsáki, 2006*; *Ji and Wilson, 2007*; *Schmidt et al., 2009*). Is such a synchronous change in excitability necessary for the generation of theta and SPW sequences? Our previous data supports this hypothesis since theta sequences were eliminated along with the waves of excitability – theta oscillations – after MS inactivation.

To further address this question, we reasoned that if the excitatory drive is necessary to generate sequences, we should not see replay of place field and episode field sequences between SPWs. To test this hypothesis we searched for sequences with a similar composition to the place field and episode field sequences throughout the entire time period of reward except for the periods when SPWs occurred (*Figure 5D*, speed <3 cm/s, excluding SPW events, see Materials and methods). However, we did not find such sequences with a probability higher than chance within a collection of 1822 time periods with sufficient neuronal activity to be able to detect such sequences (*Figure 5E*, episode field sequence replay z-score= −0.2 ± 0.23; place field sequence replay z-score= −0.15 ± 0.25; mean: 240 ± 49 time periods per recording; see Materials and methods). This data shows that both theta sequences and SPW sequences are generated exclusively at the times periods of temporally increased population excitability.

## Discussion

SPW sequences were surmised to be necessary for memory, route planning, and learning (*Buzsáki, 1989*, *2015*; *Girardeau et al., 2009*; *Dupret et al., 2010*; *Ego-Stengel and Wilson, 2010*; *Jadhav et al., 2012*; *Pfeiffer and Foster, 2013*), in part, because selective perturbation of SPWs in untrained animals impaired the speed with which animals learned to perform an episodic memory task. This impairment was observed even though other internal sequences such as theta sequences were preserved (*Jadhav et al., 2012*). Therefore, SPW sequences are considered key physiological phenomena supporting rodent navigation and memory storage. Can we thus assert that the presence of SPW sequences predicts memory performance of an animal? Based on our data, the answer

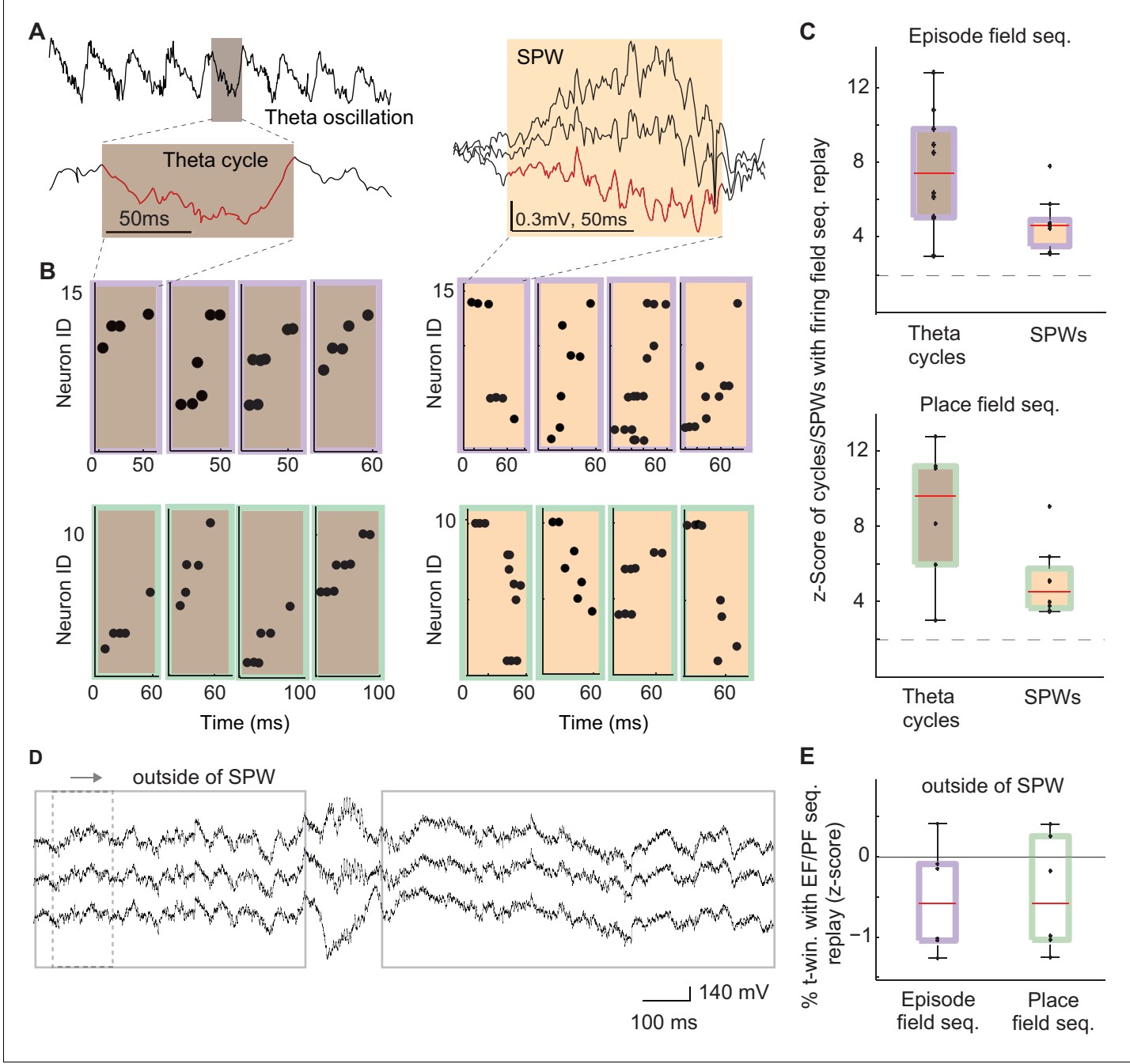

**Figure 5.** The frequency of theta and SPW sequences during and in between synchronized waves of excitability. (**A**) Left: example electrode-signal with theta oscillations. One theta cycle is magnified (red). Right: example SPW event. (**B**) Left: example sequences generated during individual cycles of theta oscillations that were correlated with episode field (brown/purple) and place field (brown/green) sequences. Right: example SPW sequences correlated with episode field (orange/purple) and place field (orange /green) sequences. (**C**) Normalized percentage of theta cycles and SPW events that contained sequences significantly correlated with episode field (top) and place field (bottom) sequences. Dash line: 2SD. (**D**) The method used to detect replay in between SPW events. (**E**) Normalized percentage of time windows that contained sequences significantly correlated with episode field (purple) and place field (green) sequences in between SPWs.

is no: SPW sequences were fully preserved in animals that could not perform an episodic memory task. It should be noted that the impact of the MS inactivation is rather broad as it affects theta oscillations, theta sequences, hippocampal cholinergic input and spatial selectivity of grid cells. Given our data and the results of others (*Girardeau et al., 2009*; *Dupret et al., 2010*; *Ego-Stenget and Wilson, 2010*; *Jadhav et al., 2012*), it is possible that SPW sequences are required only for learning but are not necessary once the environment and the task have become familiar. In contrast, theta oscillations, theta sequences, and episode field sequences seem to be necessary during learning as well as during performance of well-trained episodic memory tasks (*Landfield et al., 1972*; *Berry and Thompson, 1978*; *Winson, 1978*; *Mitchell et al., 1982*; *Mizumori et al., 1990*; *M'Harzi and Jarrard, 1992*; *Robbe et al., 2006*; *Wang et al., 2015*). Since general spatial memory of animals was not impaired after MS inactivation, as evidenced by the fact animals understood the location of the water ports, did not re-explore the maze as if it were a novel environment and remembered the procedure of the task, it is possible that neither theta sequences nor episode field sequences are necessary to make use of already established long-term spatial memory (*Hepler et al., 1985*; *Shen et al., 1996*). In contrast, theta and episode field sequences might be necessary for the instantaneous formation and recall of episodic memories such as which arm the animal just visited.

The hippocampal network was able to generate SPW sequences that were highly similar to episode field sequences even after the latter sequences were abolished. We wondered whether there is a mechanism that helped CA1 to 'read out' the network structure during SPWs that was perhaps eliminated by MS inactivation during theta state. Our data shows that SPW sequences were selectively generated when the excitability of a large portion of a network was temporarily increased, likely due to synchronized excitatory inputs from CA3 (*Buzsáki, 1986*; *Csicsvari et al., 2000*). Similarly, others have shown that optical stimulation of CA1 region triggers SPW-like events that are accompanied by sequential firing of neurons (*Stark et al., 2015*). Thus, could such a synchronized excitability increase be required to generate theta sequences? One of the effects of the medial septum inactivation was the elimination of synchronized changes of excitability during running, namely theta oscillations (*Green and Arduini, 1954*; *Petsche et al., 1962*; *Stumpf et al., 1962*). Therefore, we propose that the loss of theta sequences may be due to the lack of such excitability changes after MS inactivation. Overall, we hypothesize that either intermittent or rhythmic synchronized excitability changes are necessary to activate internally generated sequences and thus support memory (in case of theta sequences) and learning (in case of SPWs and theta sequences).

How could such synchronized changes in excitability enable the generation of internal sequences? First, the membrane potential of neurons is 2–10 millivolts below the spiking threshold during both theta and SPW brain states (*Ylinen et al., 1995*; *Harvey et al., 2009*; *Bittner et al., 2015*). The incoming wave of excitation arriving from CA3 and/or EC might synchronously increase the membrane potential among a large number of neurons, bringing all neurons closer to spiking at about the same time. The first neurons that fire during this wave of excitability likely activate local feed-forward inhibition and thus start to silence their 'competitor' principal neurons and dominate the activity of the network (*Csicsvari et al., 1999*; *Wang et al., 2015*; *Romani and Tsodyks, 2015*). Only after the competitor principal cells are silenced would these neurons be able to reliably activate their own post-synaptic principal neurons and thus trigger a sequential firing in the network.

Our data indicates that no theta and SPW sequences were generated without excitatory waves even though there are many theoretical models that can produce sequences in the absence of a broad synchronizing input. However, these models usually utilize an alternative to such an excitatory wave, such as prominent sensory cue inputs that strongly stimulate a subset of neurons at the onset of each sequence (*Tsodyks et al., 1996*; *Lisman et al., 2005*), selective amplification of already strong synaptic connections that promotes activation of a subset of cells (*Fiete et al., 2010*), or very strong connectivity, which gives rise to a distinct region of high activity that perpetually activates itself due to asymmetry of connections, dynamic thresholds, or dynamic strength of synapses (*Kleinfeld, 1986*; *Sampolinsky and Kanter, 1986*; *Horn and Usher, 1989*; *Hopfield, 2010*; *Itskov et al., 2011*). We propose that synchronized waves of excitability are required for the generation of sequences in networks with no sensory inputs, with relatively weak connections and with relatively sparse activity (*Wang et al., 2015*; *Romani and Tsodyks, 2015*). Without such synchronized waves, the activity of the network is dominated by non-specific feed-forward inhibition that prevents activation of specific sequences.

Different anatomical regions control the synchronized changes of excitability in CA1 during different brain states. SPWs are induced by inputs from the CA3 hippocampal sub-field (*Buzsáki, 1986*; *Csicsvari et al., 2000*) while theta waves are controlled by inputs from CA3, MS, and the entorhinal cortex (*Kocsis et al., 1999*). However, these synchronizing inputs still might recruit highly similar subpopulations of cells (*Varga et al., 2012*) and thus enable generation of highly similar sequences.

It has been shown that internal sequences are locked to individual cycles of rhythmic excitatory waves (oscillations) in other brain structures (*Ji and Wilson, 2007*; *Luczak et al., 2009*; *Peyrache et al., 2009*; *Havenith et al., 2011*; *Harvey et al., 2012*; *Xu et al., 2012*; *Carrillo-Reid et al., 2015*; *Markowitz et al., 2015*; *Mello et al., 2015*). Thus, the same mechanism for sequence generation might be utilized not only in the hippocampus but also in other brain regions.

Overall, we show that SPW sequences were fully preserved even in animals with impaired episodic memory and suggest that the role of SPWs might be limited to the conditions under which the animal is learning a new environment or a new aspect of a task. In contrast, the loss of theta oscillations and the associated loss of theta sequences appear to hinder both learning as well as performance of an episodic memory task. In addition, we suggest that generation of internal hippocampal sequences such as theta and SPW sequences requires not only a network substrate structured due to past experiences but also synchronized changes in the excitability of large populations of neurons.

## Materials and methods

All procedures were approved by the Janelia Farm Research Campus Institutional Animal Care and Use Committee. All data utilized in this study were previously used in *Wang et al. (2015)*. All experimental procedures were described in detail in *Wang et al. (2015)*. Here we describe data analysis methods specific for this work. No statistical methods were used to predetermine sample sizes, but our sample sizes are similar to those reported in previous publications. We used non-parametric statistical methods to reduce the effect of the small sample size.

### SPW detection

SPW events were identified based on their signature shape: positive-going LFP in the deep CA1 pyramidal layer and negative-going LFP in the superficial CA1 pyramidal layer (*Figure 1b*). Specifically, we calculated the difference between the raw LFP traces recorded from the deep and superficial CA1 and subtracted the running average calculated with a 0.5 s time window. Then we smoothed the resulting trace with a Gaussian kernal (SD = 5ms) and computed z-score. We detected individual SPW events by thresholding the smoothed trace at 4 SD and refined the start and the end time point of each event by thresholding at 1.25 SD. Finally, we extended the start and end times of each detected event in case the mean firing rate of the multi-unit activity was above 70Hz. All SPW events that were shorter than 25 ms or longer than 250 ms were excluded.

### SPW sequence detection

All spikes generated by any reliably identified neuron (spikes belonging to well isolated clusters defined as being more than 20 SD from the noise cluster) that were generated within the time period of a SPW were identified. We subselected all SPW events with more than 5 active neurons. Then we adapted the 'score' method (*Gupta et al., 2010*, *2012*; *Wang et al., 2015*) in order to identify SPW sequences that followed approximately the same order as place and episode field sequences (see Step 1 and 2 described below). In addition, we also used 'global statistics' in order to eliminate those recordings in which in the pre-muscimol session the detected percentage of significant sequences was lower than or equal to shuffled data (*Wang et al., 2015*). This step enabled us to eliminate recordings that did not have sufficient frequency of sequences replay before the manipulation. Step 1: in this step we derived a sequence score for each SPW event. The sequence score described how similar the order of spikes within that SPW event was to the order of place/episode fields of the same neurons. Specifically, for each pair of spikes we added '+1' to the sequence score, if the order of spikes was the same as the order of the episode/place cell sequence. Otherwise, we added '−1'. The sum of scores across all pairs of spikes within the time window determined the sequence score.

Step 2: in this step we determined whether a specific sequence was similar to a template place/episode field sequence above chance level by comparing the calculated sequence score to the distribution of scores obtained from randomly shuffled SPW sequences. We shuffled spikes within the SPW events by randomly reassigning the identity of each spike to a neuron which was active in the current SPW event, while preserving individual spike timing and re-calculated the sequence score as described in Step 1. This method shuffled both the identity of the cells and the order of spikes. The shuffling was repeated 10,000 times. If the original sequence score was > 97.5% or < 2.5% of the shuffled sequence scores, then the sequence was considered a significant sequence. This means only sequences with p strictly < 0.05 (including both, positive and negative replay) were considered significant sequences.

Finally, we performed global statistics on each recording by asking whether in the pre-muscimol session the percentage of significant sequences we found in our data was larger than 95% of the percentages of significant sequences we found among the entire population of shuffled sequences generated in steps 1 and 2 (Sup. Fig. 17 in *Wang et al., 2015*). This analysis partially compensated the differences in the significant sequence detection due to the choice of more liberal or restrictive shuffling procedures.

In the end, to account for the fact that different shuffling procedures result in different percentage of significant sequences and different chance levels, we chose to plot results as z-scores rather than as percentages. This step controlled for the specific choice of a more liberal or restrictive shuffling procedure:

$$\text{z-score} = (\% \text{ of SPW sequences observed in the data} - \text{mean}\% \text{ of SPW sequences observed in the shuffled data}) / (\text{STD}\% \text{ of SPW sequences observed in the shuffled data})$$

Since pyramidal neurons can burst during SPWs, shuffling the cell identity of each spike independently can break the spike statistics and lower the threshold for significant sequences. Therefore, we re-ran our analysis after we replaced each burst with the first spike of the burst. However, we obtained the same results as before.

Finally, we also re-ran the analysis with a less permissive shuffling procedures: first, we only shuffled the identity of cells within each SPW event and, second, we shuffled the identity of cells and the times of all spikes while preserving the inter-spike intervals among the spikes generated by individual cells. We followed each of these shuffling procedures with global statistics as described above. Both shuffling procedures resulted in the same overall results, that is, the replay of both place and episode field sequences was present before and after septal inactivation.

## Sequence replay search outside of SPW events (*Figure 5e*)

First, we detected time periods during which an animal was in one of the reward areas, was moving with the speed < 3 cm/s, and was not generating any SPW events (detected as described above). We slid a time window of the length of an average SPW event along these isolated reward periods. Once we found a time period with at least the minimal number of active neurons and the minimal number of spikes that enabled us to reach a significance level smaller than 0.05, we used the step 1 and 2 described in 'SPW sequence detection' section to detected sequences that were similar to one of the template sequences. Note, in this analysis we skipped over those time periods in which we previously detected SPW events.

## Acknowledgements

We thank G. Buzsáki, S. Romani, V. Itskov, A. Lee and B. Lustig for critical discussions and advice. We thank R.L. Wright for editing suggestions. This work was supported by the Howard Hughes Medical Institute (EP).

## Additional information

### Funding

| Funder | Author |
|---|---|
| Howard Hughes Medical Institute | Wang Yingxue<br>Zachary Roth<br>Eva Pastakova |

The funders had no role in study design, data collection and interpretation, or the decision to submit the work for publication.

### Author contributions

YW, EP, Conception and design, Acquisition of data, Analysis and interpretation of data, Drafting or revising the article; ZR, Conception and design, Analysis and interpretation of data, Drafting or revising the article

### Author ORCIDs

Eva Pastalkova, http://orcid.org/0000-0001-5518-9590

### Ethics

Animal experimentation: All of the animals were handled according to approved institutional animal care and use committee (IACUC) of Janelia Research Campus, HHMI: protocols 10-59 and #13-96.

## Additional files

### Major datasets

The following dataset was generated:

| Author(s) | Year | Dataset title | Dataset URL | Database, license, and accessibility information |
|---|---|---|---|---|
| Yingxue Wang, Zachary Roth, Eva Pastalkova | 2017 | Data from: Synchronized excitability in a network enables generation of internal neuronal sequences | http://dx.doi.org/10.5061/dryad.j021s | Available at Dryad Digital Repository under a CC0 Public Domain Dedication |

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
