## [Decision Letter]

[Editors’ note: a previous version of this study was rejected after peer review, but the authors submitted for reconsideration. The first decision letter after peer review is shown below.]

Thank you for submitting your work entitled "Synchronized excitability in a network enables generation of internal neuronal sequences" for consideration by *eLife*. Your article has been reviewed by three peer reviewers, and the evaluation has been overseen by a Reviewing Editor and Eve Marder as the Senior Editor. The reviewers have opted to remain anonymous.

Our decision has been reached after consultation between the reviewers. Based on these discussions and the individual reviews below, we regret to inform you that your work will not be considered further for publication in *eLife*.

Overall the reviewers found the study quite interesting and potentially important. However, there were also major concerns, especially about the method for detecting replay events and that ripple associated replay under MS inactivation alone is a rather expected result. Detailed comments are provided below.

*Reviewer #1:*

Yingxue, Roth, and Pastakova recorded spiking and local field potential data from CA1 in rats as they traversed an alternation task. The task required the animal to run on a 'running wheel' during the delay period. The authors have several main findings. First, they report that episode field sequences are replayed during sharp wave ripples with a similar consistency as place field sequences. Second, they show that muscimol inactivation of the medial septum (which reduces theta oscillations) did not block the occurrence of sharp wave ripples – in fact, the number of ripples increased. Third, they show that replay of place field sequences and episode field sequences were preserved during ripples in the muscimol condition (despite the disruption of episode field sequences during muscimol inactivation). Finally, the authors show that replay events of place field and episode field sequences are generally restricted to sharp wave ripple events.

In general I find the paper will be interesting to many, but I feel that the conclusions drawn by the authors about the role of sharp wave ripples in memory is grandiose. The authors point to prior reports that selective elimination of sharp wave ripples disrupt memory performance and consolidation. They then frame muscimol inactivation as a method to test memory while preserving sharp wave ripples to test whether sharp wave ripples are sufficient to support memory. The authors state that since muscimol inactivation disrupts memory performance, then sharp-wave ripple activity is not sufficient to support memory. I cannot understand why this is an interesting conclusion – and I feel that the authors have reached too far here. Muscimol inactivation of the septum does many things to hippocampal and entorhinal physiology including the reduction or elimination of 1) theta oscillations, 2) cholinergic input, 3) speed modulated input, 4) theta sequences, 5) episode/time fields, 6) spatial selectivity of grid cell input, 7) theta cycle skipping of HD cells. With so much disrupted it’s almost meaningless to say that the one or two things that remain intact are not sufficient for memory performance. An analogy would be to remove 90% of a car engine and conclude that the transmission alone isn't enough for the car to operate. At a minimum I would suggest the authors discuss this point in detail.

The authors also conclude that sharp wave ripples might then be needed for learning. Using their own logic, do the authors predict that learning will be intact during septal inactivation? This seems incredibly unlikely. While I agree that sharp wave ripples may be needed for learning (didn't Wilson et al. 1994 say this?) I don't think the data presented here addresses this point at all.

The most interesting parts of this paper are that 1) there is replay of episode fields, 2) replay in conserved during septal inactivation, 3) replay of episode fields is conserved even though episode fields are disrupted during septal inactivation. I think the authors need to focus on these points and probably forget about their discussion of memory. On a positive note, I did appreciate the discussion on how a "wave of excitability" could generate a sequence.

*Reviewer #2:*

This manuscript observed place cell sequences during sharp wave ripples (SPW-Rs) under the suppression of the medial septum during a delayed alternation task. Septal inhibition increased the incidence of SPW-Rs and increased the error rates of the animal (as reported in their previous paper). Therefore, the authors suggest that sequence reactivation during SPW-Rs is not sufficient for the animal to perform the delayed alternation task. The work is potentially significant, however, they need to address several technical questions.

1) According to their previous paper, septal inhibition increased the error rate but the animal still showed a bias to choose the correct arm. Therefore, it is important to show that, during error trials, sequences reactivation can still be seen.

2) Is there a correlation between the error rate observed in a session and quality of sequence reactivation?

3) Although SPW-R may not be needed for memory recall, it is possible that they may reflect the arm choice of the animal. Therefore, it is needed to be quantified whether reactivated sequences reflect preferentially the past or future choice of the animal under septal inhibition. They report these for the before inactivation case but their description is confusing. This data needs to be described better and I think it would also deserve a separate figure considering that it may not agree with previous work.

4) SPW-Rs were not detected the 'usual way', in fact they detected high synchrony periods and the Methods section does explain how it was verified whether indeed ripples were present during these periods. According to Figure 4 no sequences were detected outside SPW-Rs. Does it mean that high synchrony periods with low ripple power did not contain sequences? Again, if this was the case, it is surprising and it does not agree with past work that usually shows a 10-20% correspondence between SPW-Rs and high synchrony periods.

*Reviewer #3:*

In this study, the authors examine the effect of inactivation of the medial septum (MS) on the generation of sequential patterns of activity across ensembles of hippocampal CA1 neurons in the context of a dual arm alternation memory task, modified to include a running wheel. This work builds on prior work by the same group (Wang et al., 2015) and it represents new analyses performed on this same data set. The main findings reported by the authors include 1) the observation that episode field sequences are "replayed" during sharp wave ripple LFP events at rate that is higher than would be expected by chance and similar to the rate of place field sequence replay. 2) Inactivation of MS using muscimol does not abolish SPW replay of event/place field sequences, in fact it increases the occurrence of these replay events. However, I believe that the statistics employed to detect replay events are flawed, and thus I do not believe that the data supports the main claims of the paper. I outline the nature of my concerns below.

Though one has to dig through the previous paper to understand the methods for detecting sequences of firing on small time scales that are similar to sequences of firing on longer timescales, I believe I understand the analysis steps taken by the authors, and I believe they are problematic. The authors first detect episode/place field sequences as described in Wang 2015. They derive from these sequences an ordered sequence of firing peaks across cells. To detect replay events, they then compute a sequence similarity score by comparing each pair of spikes recorded in different cells within a time window, adding 1 to a sequence score if the order of those two spikes is the same as the order of the peak firing rates in the episode/place field sequence, and adding -1 otherwise. To assess significance, the authors shuffle the cell identity of each spike, maintaining spike timing, across the active cells within the epoch. They do this 10,000 times to build a null distribution of sequence scores. If the observed sequence score falls outside of 95% of the shuffle-computed sequence scores, the authors identify the sequence as being significantly similar to the episode/place field sequence.

However, it appears from the examples shown that there is often structure in the autocorrelation function of neurons' spiking, for example, neurons seem to burst in many identified sequences. By shuffling the cell identity of each spike independently, the authors break this feature of spiking statistics, inflating the significance of patterns they identify as sequences. For an extreme example, imagine two cells fire three spikes each in non-overlapping bursts, in the order found in the original episode/place field sequence. Shuffling spikes independently makes the particular observed sequence score unlikely by chance. However, if the autocorrelation function of the cells were to be preserved, that particular sequence score becomes very likely by chance, essentially those bursts of spikes would need to travel together in the reshuffling process. I think this is a huge problem for their data and conclusions.

Unfortunately, for me, since much of the claims of the paper rely on accurately identifying replay events, these issues are a non-starter for considering the validity of the authors' conclusions. The authors would need to convince me that performing their shuffling procedure in a way that maintains the structure of the autocorrelation within neurons does not change their results for me to consider the paper further.

---

## [Author Response]

[Editors’ note: the author responses to the previous round of peer review follow.]

*Reviewer #1:*

*[…] In general I find the paper will be interesting to many, but I feel that the conclusions drawn by the authors about the role of sharp wave ripples in memory is grandiose. The authors point to prior reports that selective elimination of sharp wave ripples disrupt memory performance and consolidation. They then frame muscimol inactivation as a method to test memory while preserving sharp wave ripples to test whether sharp wave ripples are sufficient to support memory. The authors state that since muscimol inactivation disrupts memory performance, then sharp-wave ripple activity is not sufficient to support memory. I cannot understand why this is an interesting conclusion – and I feel that the authors have reached too far here. Muscimol inactivation of the septum does many things to hippocampal and entorhinal physiology including the reduction or elimination of 1) theta oscillations, 2) cholinergic input, 3) speed modulated input, 4) theta sequences, 5) episode/time fields, 6) spatial selectivity of grid cell input, 7) theta cycle skipping of HD cells. With so much disrupted it’s almost meaningless to say that the one or two things that remain intact are not sufficient for memory performance. An analogy would be to remove 90% of a car engine and conclude that the transmission alone isn't enough for the car to operate. At a minimum I would suggest the authors discuss this point in detail.*

We thank the reviewer for these comments. We apologize if our conclusions came across as grandiose. We edited the text so it hopefully describes the results without over-interpretation. We also included a discussion on the widespread effect of medial septum inactivation, as suggested by the reviewer (Discussion, first paragraph).

The reviewer also posed a question why the conclusion “sharp-wave ripple activity is not sufficient to support memory” is interesting. We reformulated our conclusion to clarify this point. Specifically, as the reviewer pointed out, prior reports suggested that selective elimination of sharp-wave ripples disrupts memory performance and consolidation (Girardeau et al., 2009, Ego-Stengel and Wilson, 2010, Nokia et al., 2012, Jadhav et al., 2012). Previous studies also showed that the sequences during sharp-wave events can replay the ongoing experience (Pavlides et al., 1989, Eschenko et al., 2008, Karlsson et al., 2009, Singer et al., 2009, Dupret et al., 2010, Girardeau et al., 2014), the past experience (Karlsson and Frank, 2009), or predict animal’s future behavior (Singer at al., 2013, Pfeiffer et al., 2013). Together, the available evidence supports the hypothesis that sharp-wave replay plays an important role in memory performance (Carr et al., 2011, Girardeau and Zugaro, 2011, Sadowski et al., 2011, Csicsvari and Dupret, 2013, Buzsaki, 2015). Here we are able to supply evidence that sheds somewhat different light on the role SPW sequences play in memory. Specifically, our result dissociates the occurrence of the sharp-wave replay from memory performance: it shows that the presence of SPW sequences does not always indicate whether an animal is capable of performing a memory task or not. Despite the wide range of consequences induced by medial septum inactivation, this observation urges us to reassess the role of sharp-wave replay in learning and memory. We included these points in the Discussion (first paragraph).

*The authors also conclude that sharp wave ripples might then be needed for learning.*

We would like to clarify that we made this statement purely based on published studies (Girardeau et al., 2009, Ego-Stengel and Wilson, 2010, Nokia et al., 2012, Jadhav et al., 2012). Indeed, we could not make this statement based on the data presented in this manuscript.

*Using their own logic, do the authors predict that learning will be intact during septal inactivation? This seems incredibly unlikely. While I agree that sharp wave ripples may be needed for learning (didn't Wilson et al. 1994 say this?) I don't think the data presented here addresses this point at all.*

We believe, we might have not been clear in our Discussion and caused some misunderstanding between the reviewer and us. As we explained above, we showed that SPW sequences can exist even after performance in the memory task was impaired. Thus, our results showed that the presence of one phenomenon – sharp wave sequences – does not predict the presence of the other phenomenon – learning and/or memory. As a result, we could not predict what the memory performance of an animal might be solely based on the fact that SPW sequences were present. We now tried to clarify these points in the Discussion (first paragraph).

*The most interesting parts of this paper are that 1) there is replay of episode fields, 2) replay in conserved during septal inactivation, 3) replay of episode fields is conserved even though episode fields are disrupted during septal inactivation. I think the authors need to focus on these points and probably forget about their discussion of memory. On a positive note, I did appreciate the discussion on how a "wave of excitability" could generate a sequence.*

We thank the reviewer for this comment.

*Reviewer #2:*

*This manuscript observed place cell sequences during sharp wave ripples (SPW-Rs) under the suppression of the medial septum during a delayed alternation task. Septal inhibition increased the incidence of SPW-Rs and increased the error rates of the animal (as reported in their previous paper). Therefore, the authors suggest that sequence reactivation during SPW-Rs is not sufficient for the animal to perform the delayed alternation task. The work is potentially significant, however, they need to address several technical questions.*

*1) According to their previous paper, septal inhibition increased the error rate but the animal still showed a bias to choose the correct arm. Therefore, it is important to show that, during error trials, sequences reactivation can still be seen.*

We followed the reviewer’s suggestion and compared the frequency of significant replay during correct and error trials before and after septal inactivation (the data for place and episode cell replay were merged in this analysis):

Author response table 1.**DOI:**
http://dx.doi.org/10.7554/eLife.20697.007Mean frequency of replay ± SEM:PreMuscimolCorrect trials0.22Hz ± 0.0140.32Hz ± 0.024Error trials0.5Hz ± 0.080.85Hz ± 0.25

Mean frequency of significant sequence replay during correct and error trials before and after MS inactivation (place and episode cell data were merged).

In Figure 4 we plotted the result for each recording (round markers): mean value of SPW replay for all correct (black) and error trials (red). One can see that the replay frequency was higher after septal inactivation and that the frequency did not change during error trials. Note that the high variability of replay frequency during error trials. Also, note that, we excluded those recordings where there was no error trial in the ‘Pre’ condition.

*2) Is there a correlation between the error rate observed in a session and quality of sequence reactivation?*

We did not see any significant correlation between the performance of animals and the percentage of detected significant sequences: Pre: r = 0.34 p = 0.25; Muscimol: r = −0.19 p = 0.53; Pre & Muscimol: r = −0.1 p = 0.65

We now included these data into the Results section (Figure 4 and subsection “SPW sequences were present even after memory loss”, last paragraph).

*3) Although SPW-R may not be needed for memory recall, it is possible that they may reflect the arm choice of the animal. Therefore, it is needed to be quantified whether reactivated sequences reflect preferentially the past or future choice of the animal under septal inhibition. They report these for the before inactivation case but their description is confusing. This data needs to be described better and I think it would also deserve a separate figure considering that it may not agree with previous work.*

Following these suggestions, we included data that described the preferential reactivation of the left-arm and the right-arm place field sequences in the left and right arm reward areas during muscimol into the main text (subsection “Internal hippocampal sequences are replayed during sharp-waves”, last paragraph and subsection “SPW sequences were present even after memory loss”, last paragraph). Specifically, during muscimol, the reactivation of the left-arm place field sequence was observed in the left reward area with the probability of 56.5 ± 6.6% compared to the right reward area. The right-arm place field sequence was observed with 63 ± 8.3% probability in the left reward area compared to the right reward area.

*4) SPW-Rs were not detected the 'usual way', in fact they detected high synchrony periods and the Methods section does explain how it was verified whether indeed ripples were present during these periods. According to Figure 4 no sequences were detected outside SPW-Rs. Does it mean that high synchrony periods with low ripple power did not contain sequences? Again, if this was the case, it is surprising and it does not agree with past work that usually shows a 10-20% correspondence between SPW-Rs and high synchrony periods.*

We apologize we did not include enough details in our Methods section. We now described the SPW event detection in much greater detail (subsection “Sequence replay search outside of SPW events (Figure 5)”). We would like to note that the “high synchrony periods” detection used in Figure 4 was not the same as the SPW event detection used in the rest of the manuscript. In Figure 4, we did not try to detect high synchrony events – we slid a time window along the entire time periods animals spent in the reward areas and looked for those time windows that had sufficient number of active neurons and spikes that gave us sufficient statistical power for assessing the significance of sequences replay. The length of the sliding window was equal to the length of the average SPW event in that recording. In this analysis we actively avoided and skipped over those time periods in which SPW events were detected previously. This method is now described in the aforementioned subsection.

*Reviewer #3:*

*In this study, the authors examine the effect of inactivation of the medial septum (MS) on the generation of sequential patterns of activity across ensembles of hippocampal CA1 neurons in the context of a dual arm alternation memory task, modified to include a running wheel. This work builds on prior work by the same group (Wang et al., 2015) and it represents new analyses performed on this same data set. The main findings reported by the authors include 1) the observation that episode field sequences are "replayed" during sharp wave ripple LFP events at rate that is higher than would be expected by chance and similar to the rate of place field sequence replay. 2) Inactivation of MS using muscimol does not abolish SPW replay of event/place field sequences, in fact it increases the occurrence of these replay events. However, I believe that the statistics employed to detect replay events are flawed, and thus I do not believe that the data supports the main claims of the paper. I outline the nature of my concerns below.*

*Though one has to dig through the previous paper to understand the methods for detecting sequences of firing on small time scales that are similar to sequences of firing on longer timescales, I believe I understand the analysis steps taken by the authors, and I believe they are problematic. The authors first detect episode/place field sequences as described in Wang 2015. They derive from these sequences an ordered sequence of firing peaks across cells. To detect replay events, they then compute a sequence similarity score by comparing each pair of spikes recorded in different cells within a time window, adding 1 to a sequence score if the order of those two spikes is the same as the order of the peak firing rates in the episode/place field sequence, and adding -1 otherwise. To assess significance, the authors shuffle the cell identity of each spike, maintaining spike timing, across the active cells within the epoch. They do this 10,000 times to build a null distribution of sequence scores. If the observed sequence score falls outside of 95% of the shuffle-computed sequence scores, the authors identify the sequence as being significantly similar to the episode/place field sequence.*

We apologize for the insufficient Methods section in the originally submitted manuscript. We have now greatly expanded this section of the manuscript so that all readers can access the methods easily. We would like to confirm, we think, the reviewer indeed understood our method correctly. The only note we would like to make is that we used > 97.5% or < 2.5% as the significance threshold for detection of the positive or negative replay sequences as opposed to 95% threshold.

*However, it appears from the examples shown that there is often structure in the autocorrelation function of neurons' spiking, for example, neurons seem to burst in many identified sequences. By shuffling the cell identity of each spike independently, the authors break this feature of spiking statistics, inflating the significance of patterns they identify as sequences. For an extreme example, imagine two cells fire three spikes each in non-overlapping bursts, in the order found in the original episode/place field sequence. Shuffling spikes independently makes the particular observed sequence score unlikely by chance. However, if the autocorrelation function of the cells were to be preserved, that particular sequence score becomes very likely by chance, essentially those bursts of spikes would need to travel together in the reshuffling process. I think this is a huge problem for their data and conclusions.*

We agree with the reviewer that our shuffling procedure does not preserve the auto- correlation function of individual spike trains and as a result is more liberal than the procedure suggested by the reviewer. However, we would like to propose that we control for this fact in two ways:

1) We estimated the global significance of each recording, meaning that in addition to estimating the significance of each SPW sequence, we also asked whether the total number of detected significant SPW sequences was above the chance level in each recording. Specifically, we compared the percentage of significant sequences detected in a specific recording with the distribution of the percentage of significant sequences detected in the entire collection of shuffled sequences generated during all shuffling steps (for details, please refer to Sup. Figure 17 in Wang et al., 2015). Since our shuffling procedure that generated the population of shuffled sequences is liberal, the number of significant sequences detected in the shuffled data was proportionally higher than it would have been with a more conservative procedure. Thus, our global statistical test corrected for the more liberal shuffling method, and only included those recordings in which the percentage of significant sequences in the pre-muscimol condition passed the global statistical test into the data analysis.

2) We plotted and reported the data as z-score as opposed to the number or percentage of sequences. The z-score was calculated as follows:

z-score = (% of SPW sequences observed in the data – mean% of SPW sequences observed in the shuffled data) / STD% of SPW sequences observed in the shuffled data

This way, we accounted for the fact that different shuffling procedures (as shown below) were differently strict and resulted in different chance levels.

*Unfortunately, for me, since much of the claims of the paper rely on accurately identifying replay events, these issues are a non-starter for considering the validity of the authors' conclusions. The authors would need to convince me that performing their shuffling procedure in a way that maintains the structure of the autocorrelation within neurons does not change their results for me to consider the paper further.*

We think the use of global statistics combined with the use of z-score is an appropriate way to control for the more liberal nature of our shuffling procedure. However, following reviewer’s comment we wanted to convince ourselves that our results are not dependent on the presence of bursts and on the use of a more liberal shuffling procedure. To test this, we did the following 3 tests:

Test 1: we detected significant SPW sequences after we eliminated all spikes in each burst except for the first one. We still observed preserved SPW replay after medial septum injection (Figure 6). Following previous literature, we did not apply our global statistical test and thus, included all recordings in the plots. Also, we plotted the data as percentage of sequences with significant replay.

Author response image 1.SPW replay of episode cell (left) and place cell (right) sequences during Pre-muscimol and Muscimol recordings.In this analysis we eliminated all spikes from each burst except for the first one. We did not observe any drop in the prevalence of SPW replay during Muscimol. The chance level (dash line) is determined by the mean of the percentages of significant replay observed in the shuffled data.**DOI:**
http://dx.doi.org/10.7554/eLife.20697.008

Test 2: we changed our shuffling procedure to follow the recommendation of the reviewer: we shuffled the identity of neurons and the time stamps of spikes so that the auto-correlation function of neurons was preserved (Figure 7).

Author response image 2.SPW replay of episode cell (left) and place cell (right) sequences during Pre-muscimol and Muscimol recordings.In this analysis we shuffled the identity of cells and time stamps but preserved the autocorrelation function of each cell. The chance level (dash line) is determined by the mean of the percentages of significant replay observed in the shuffled data. We did not observe any drop in the prevalence of SPW replay during Muscimol.**DOI:**
http://dx.doi.org/10.7554/eLife.20697.009

Test 3: we modified our shuffling procedure once again and, following the recommendation of the reviewer, we shuffled only the identity of neurons (Figure 8). Again, we included all recordings, irrespective of whether the pre-muscimol conditions were considered significant by the global statistical test.

Author response image 3.SPW replay of episode cell (left) and place cell (right) sequences during Pre-muscimol and Muscimol recordings.In this analysis we shuffled the identity of neurons without altering the spike timing. The chance level (dash line) is determined by the mean of the percentages of significant replay observed in the shuffled data. We did not observe any drop in the prevalence of SPW replay during Muscimol.**DOI:**
http://dx.doi.org/10.7554/eLife.20697.010

Overall, we reanalyzed our data with 4 different shuffling procedures that ranged from liberal to very strict. As one might expect, we observed lower percentage of SPW sequences with significant replay when we used more strict methods. However, we were able to reproduce our original findingwith all of these methods, that is, we showed that there was no significant drop in SPW replay between pre-muscimol and muscimol conditions. Overall, all these results support the conclusion in our manuscript. We thank the reviewer again for motivating us to improve our manuscript. We hope these results would help the reviewer to further consider our manuscript.